

# Efficacy and safety of etrolizumab in the treatment of inflammatory bowel disease: a meta-analysis

Yong gang Dai[1,2], Dajuan Sun[1], Jiahui Liu[1], Xiunan Wei[1], Lili Chi[1] and Hongya Wang[2]

[1] Shandong University of Traditional Chinese Medicine, Shandong, China
[2] Shandong Provincial Third Hospital, Shandong, China

## ABSTRACT

**Background.** To explore the efficacy and safety of etrolizumab in treating inflammatory bowel disease (IBD) through meta-analysis.

**Method.** A comprehensive exploration encompassed randomized controlled trials examining the efficacy of etrolizumab in treating IBD across PubMed, Embase, Cochrane library, and Web of Science, with a search deadline of 1 December 2023. Quality assessment leaned on the Cochrane manual's risk-of-bias evaluation, while Stata 15 undertook the data analysis.

**Result.** Five randomized controlled studies involving 1682 individuals were finally included, Meta-analysis results suggested that compared with placebo, etrolizumab could improve clinical response (RR = 1.26, 95% CI [1.04–1.51]), clinical remission (RR = 1.26, 95% CI [1.04–1.51]) in IBD patients. Endoscopic alleviate (RR = 2.10, 95% CI [1.56–2.82]), endoscopic improvement (RR = 2.10, 95% CI [1.56–2.82]), endoscopic remission (RR = 2.10, 95% CI [1.56–2.82]), Endoscopic improvement (RR = 1.56, 95% CI [1.30–1.89]), histological remission (RR = 1.62, 95% CI [1.26–2.08]), and did not increase any adverse events (RR = 0.95, 95% CI [0.90–1.01]) and serious adverse events (RR = 0.94, 95% CI [0.68–1.31]).

**Conclusion.** According to our current study, etrolizumab is a promising drug in IBD.

## INTRODUCTION

Inflammatory bowel disease (IBD) is a chronic, idiopathic inflammatory condition that can affect all parts of the digestive tract, characterized by mucosal immune dysregulation and recurrent bouts of intestinal inflammation (*Bisgaard et al., 2022*; *Sasson et al., 2021*). Ulcerative colitis (UC) and Crohn's disease (CD) represent the two expression types of this condition, distinguished by the location and depth of inflammation, closely associated with genetic, immune, lifestyle, and environmental factors (*Agrawal et al., 2021*; *Plevris & Lees, 2022*; *Welz & Aden, 2023*). Although CD and UC are different diseases, from the point of view of disease occurrence, both UC and CD belong to autoimmune diseases, and their clinical manifestations are similar, so we can analyze them together. Epidemiological

Corresponding authors
Lili Chi, chililiyl@163.com
Hongya Wang, wangya828qing@163.com

studies reveal the highest prevalence of IBD in Europe and North America, with a rapid rise in incidence in emerging industrialized nations (*Xu et al., 2021*). Individuals and their descendants migrating from regions with low IBD prevalence (such as the Middle East and South Asia) to areas with high prevalence exhibit increased susceptibility to IBD. However, the precise causes and mechanisms underlying IBD remain unclear. Inflammation and oxidative stress are generally perceived as key mechanisms in IBD pathogenesis (*Barbieri, 2021*; *Zhao et al., 2021*). Clinically, treatments for IBD mainly include drugs like 5-aminosalicylic acid, corticosteroids, immunosuppressants (such as azathioprine), and biologics (like anti-TNF-[3] agents, anti-integrins, and anti-cytokine antibodies) (*Baumgart & Le Berre, 2021*). Unfortunately, these medications merely offer symptomatic relief without curing the disease and often lead to noticeable adverse effects such as anemia, liver and kidney dysfunction, leukopenia, cataracts, osteoporosis, malignancies, immunosuppression, and an increased risk of opportunistic infections (*Hadji & Bouchemal, 2022*). Some of these adverse effects are irreversible. Moreover, research indicates that early surgery and the use of immunosuppressants fail to prevent the tendency for reoperation and disease disability in Crohn's disease patients (*Ouyang, Zhao & Wang, 2023*). Hence, there is an urgent need to discover safe and effective therapies for IBD.

In recent years, anti-integrins have been used in therapy and have shown promise (*Solitano et al., 2021*). Natalizumab, Vedolizumab and etrolizumab are part of this class of drugs. Natalizumab is a monoclonal antibody targeting the α4 integrin, which is rarely used nowadays due to safety concerns (*Gordon et al., 2002*). Vedolizumab is a selective antibody targeting the α4β7 integrin, which plays an important role in intestinal Leukocytes play an important role in the migration of leukocytes to the intestine (*Pouillon, Vermeire & Bossuyt, 2019*). Etrolizumab is a humanized monoclonal IgG1 antibody directed against the β7 subunit of the heterodimeric integrins α4β7 and αEβ7. α4β7 integrin is a key mediator of leukocyte infiltration in the gastrointestinal tract by interacting with MAdCAM-1 on the vascular endothelium of mucosal tissues (*Lichnog et al., 2019*; *Makker & Hommes, 2016*). However, the efficacy and safety of etrolizumab for IBD are still controversial (*Fiorino, Gilardi & Danese, 2016*), so we hope to resolve these controversies with this study and provide new options for clinical patient treatment.

## METHOD

The systematic review described herein was accepted by the online PROSPERO international prospective register of systematic reviews (*Page et al., 2021*) of the National Institute for Health Research (CRD42023494132).

### Inclusion and exclusion criteria

The included population met the diagnostic criteria for inflammatory bowel disease (*Watermeyer et al., 2022*). Etrolizumab was used in the experimental group and placebo was used in the control group, and the primary outcome were clinical remission (defined as (Mayo Clinic Score) MCS of ≤2); clinical response (3-point decrease and 30% reduction in MCS and 1-point decrease); endoscopic remission (defined as Mayo endoscopic sub score of 0); endoscopic improvement (defined as Mayo endoscopic sub score of ≤1); and

the secondary outcome were histological remission (defined as Nancy histological index [NHI] of ≤1 among patients with histological inflammation at baseline); adverse events, the randomized controlled trial was included in this study.

Conference abstracts, meta-analyses, systematic reviews, animal experiments, Full text is not available and case reports will be considered for exclusion.

## Literature retrieval

Randomized controlled trials on etrolizumab for inflammatory bowel disease were searched in PubMed, Embase, Cochrane Library, Web of science, with a search deadline of 1 December 2023, using the mesh word combined with a free word: etrolizumab inflammatory bowel disease. Detailed search strategies are provided in Supplemental Information 1.

## Data extract

Two authors (DYG and SDJ) rigorously screened the literature based on predetermined inclusion and exclusion criteria. In case of any disagreement, they resolved it through discussion or sought the opinion of a third person (WHY) to negotiate and reach consensus. Information extracted from the included studies included the following key details: authors, year, country, sample size, gender, mean age, Type of disease, intervention, and outcome.

## Grade of evidence

To determine the quality of our results, we selected the Graded Recommendations Assessment Development and Evaluation (GRADE) system to evaluate the evidence (*Atkins et al., 2004*) for methodological quality. We considered five factors that could reduce the quality of the evidence, including study limitations, inconsistent findings, inconclusive direct evidence, inaccurate or wide confidence intervals, and publication bias. In addition, three factors that could reduce the quality of evidence were reviewed, namely effect size, possible confounding factors, and dose–effect relationships. A comprehensive description of the quality of evidence for each parameter data is provided (Table S1).

## Included studies' risk of bias

Two investigators (DYG and SDJ) independently assessed the risk of bias as low, unclear, or high using the Cochrane Collaboration's tools (*Higgins et al., 2011*). If there was any disagreement, a third person (WHY) was consulted to reach consensus. The assessment included seven areas: generation of randomized sequences, allocation concealment, blinding of implementers and participants, blinding of outcome assessors, completeness of outcome data, selective reporting of study results, and other potential sources of bias.

## Data analysis

The collected data were statistically analyzed using Stata 15.0 software (Stata Corp, College Station, TX, USA). Heterogeneity between included studies was assessed using $I2$ values or Q-statistics. $I2$ values of 0%, 25%, 50%, and 75% indicated no heterogeneity, low heterogeneity, moderate heterogeneity, and high heterogeneity, respectively. If the $I2$ value was equal to or greater than 50%, a sensitivity analysis was performed to explore potential sources of heterogeneity. If heterogeneity was less than 50 per cent, analyses were

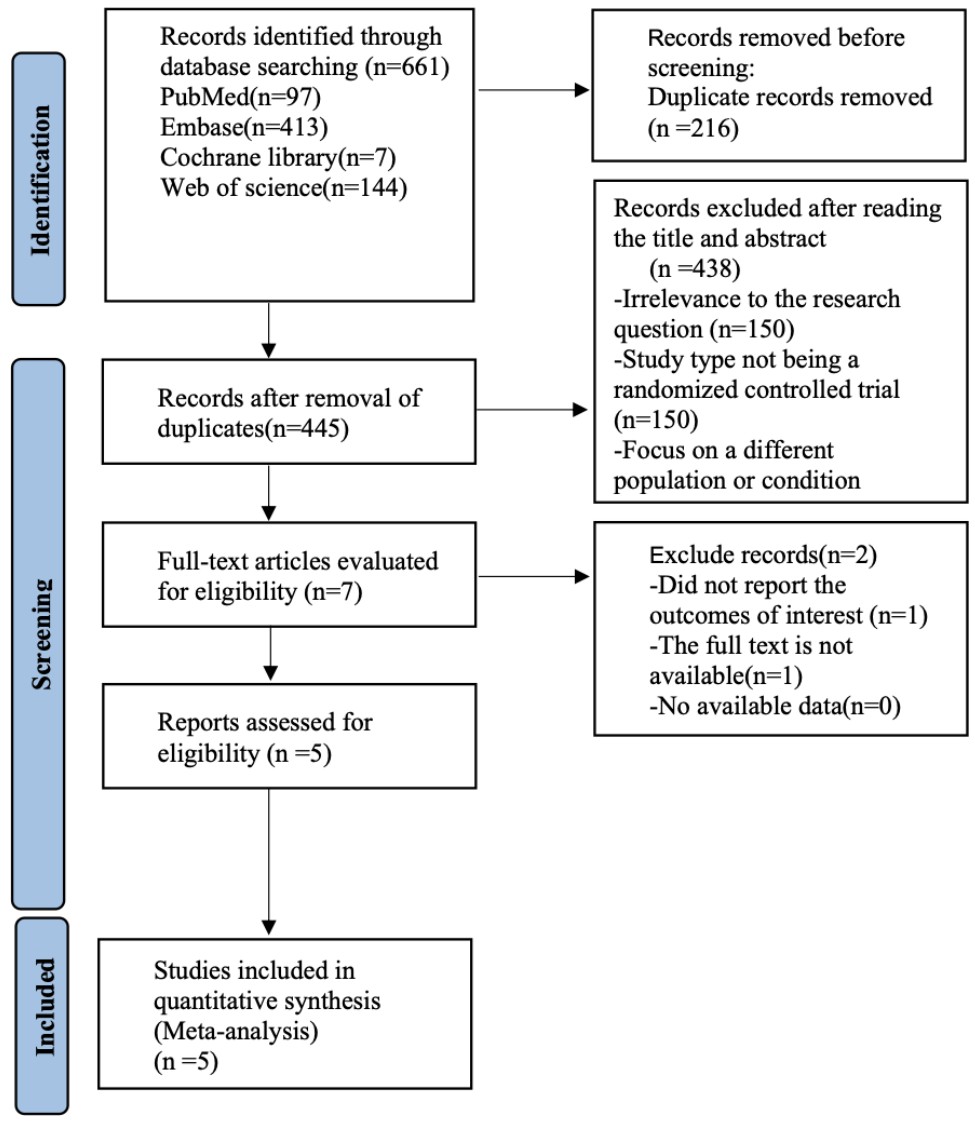

**Figure 1** Prisma flow chart.

conducted using a fixed-effects model. Risk ratio (RR) and 95% confidence interval (CI) for dichotomous variables. In addition, random effects model and Egger's test were used to assess publication bias.

## RESULT

### Study selection

Figure 1 shows our literature search process, which initially retrieved 661 documents, removed 216 duplicates, removed 438 articles by reading titles and abstracts, removed two papers by reading the full text, and finally included five (*Peyrin-Biroulet et al., 2022*; *Rubin et al., 2022*; *Sandborn et al., 2023*; *Vermeire et al., 2022*; *Vermeire et al., 2014*) randomized controlled trials for analysis.

### Basic characteristics and risk of bias of the included studies

Five randomized controlled studies involving 1,682 individuals were finally included, in four articles (*Peyrin-Biroulet et al., 2022*; *Rubin et al., 2022*; *Vermeire et al., 2022*; *Vermeire et al., 2014*) for ulcerative colitis and one (*Sandborn et al., 2023*) for Crohn's disease, doses of etrolizumab ranged from 100 to 300 mg. Baseline characteristics are shown in Table 1. The five included studies clearly accounted for the method of randomization used, and the risk of bias results are shown in Figures S2–S3.

### Result of meta-analysis
#### *Clinical response*
Five articles were divided into seven trials mentioning clinical response and the test of heterogeneity was ($I2 = 26.1\%$, $P = 0.247$), so the data were analyzed by using the fixed effect model and the analysis results (Fig. 2) suggested that compared with placebo, etrolizumab was able to improve IBD patients' clinical response (RR = 1.26, 95% CI [1.04–1.51]), and the difference was statistically significant.

### Clinical remission

Three articles were divided into five trials mentioning clinical remission and the test of heterogeneity was ($I2 = 26.1\%$, $P = 0.247$), so the data were analyzed by using the fixed effect model and the analysis results (Fig. 3) suggested that compared with placebo, etrolizumab was able to improve IBD patients' clinical remission (RR = 1.26, 95% CI [1.04–1.51]), and the difference was statistically significant.

### Endoscopic remission

Five articles were divided into seven trials mentioning endoscopic remission and the test of heterogeneity was ($I2 = 0\%$, $P = 0.826$), so the data were analyzed by using the fixed effect model and the analysis results (Fig. 4) suggested that compared with placebo, etrolizumab was able to improve IBD patients' endoscopic remission (RR = 2.10, 95% CI [1.56–2.82]), and the difference was statistically significant.

### Endoscopic improvement

Four articles were divided into five trials mentioning endoscopic improvement and the test of heterogeneity was ($I2 = 0\%$, $P = 0.556$), so the data were analyzed by using the fixed effect model and the analysis results (Fig. 5) suggested that compared with placebo, etrolizumab was able to improve IBD patients' endoscopic improvement (RR = 1.56, 95% CI [1.30–1.89]), and the difference was statistically significant.

### Adverse event

Five articles were divided into seven trials mentioned adverse events (including any adverse events and serious adverse events), for any adverse events, the heterogeneity test ($I2 = 43.2\%$, $P = 0.103$), so the data were analyzed by using the fixed effect model, and the results of the analysis (Fig. 6) suggested that compared with placebo Compared with placebo, etrolizumab was not statistically significant for any adverse events (RR = 0.95, 95% CI [0.90–1.01]); for serious adverse events, heterogeneity test ($I2 = 0\%$, $P = 0.886$), so the

**Table 1  Baseline characteristics.**

| Study | Year | Country | Race | Sample size | | Gender (M/F) | Mean age (years) | | Types of disease | Disease severity | Comorbidity | Medications | Intervention | | Outcome |
|---|---|---|---|---|---|---|---|---|---|---|---|---|---|---|---|
| | | | | EG | CG | | EG | CG | | | | | EG | CG | |
| Peyrin | 2022 | France | White and Asian | 384 | 95 | 278/201 | 39 | 36 | ulcerative colitis | moderately to severely | NR | Corticosteroid and immunosuppressant use | Subcutaneous etrolizumab 105 mg once every 4 weeks | Placebo | F1; F2; F3; F4; F5; F6 |
| Rubin | 2022 | USA | NR | 144 | 72 | 113/103 (HIBISCUS I) | 36.5 | 36 | Ulcerative colitis | Moderately to severely | NR | Corticosteroid and Immunosuppressant use | Subcutaneous etrolizumab 105 mg once every 4 weeks | Placebo | F1; F2; F3; F4; F5; F6 |
| | | | | 143 | 72 | 122/93 (HIBISCUS II) | 39 | 36.5 | | Moderately to severely | NR | Corticosteroid and immunosuppressant use | | | |
| Sandborn | 2023 | USA | White | 217 | 217 | 218/216 | 38.8 | 37.9 | Crohn's disease | Moderately to severely | NR | Corticosteroid and immunosuppressant and anti-TNF use | Subcutaneous etrolizumab 105 mg once every 4 weeks | Placebo | F1; F3; F4; F5; |
| Vermeire | 2022 | Belgium | White | 108 | 106 | 112/102 | 36 | 38 | Ulcerative colitis | Moderately to severely | NR | Corticosteroid and immunosuppressant use | Subcutaneous etrolizumab 105 mg once every 4 weeks | Placebo | F1; F3; F4; F5; F6 |
| Vermeire | 2014 | Belgium | White | 81 | 43 | 71/63 | 42 | 37.5 | Ulcerative colitis | Moderately to severely | NR | Corticosteroid and Immunosuppressant use | Subcutaneous etrolizumab 100/300 mg once every 4 weeks | Placebo | F1; F2; F3; F5 |

**Notes.**

EG, experimental group; CG, Control group; M/F, Male/Female; F1, clinical response; F2, clinical remission; F3, endoscopic remission; F4, endoscopic improvement; F5, adverse events; F6, Histological remission.

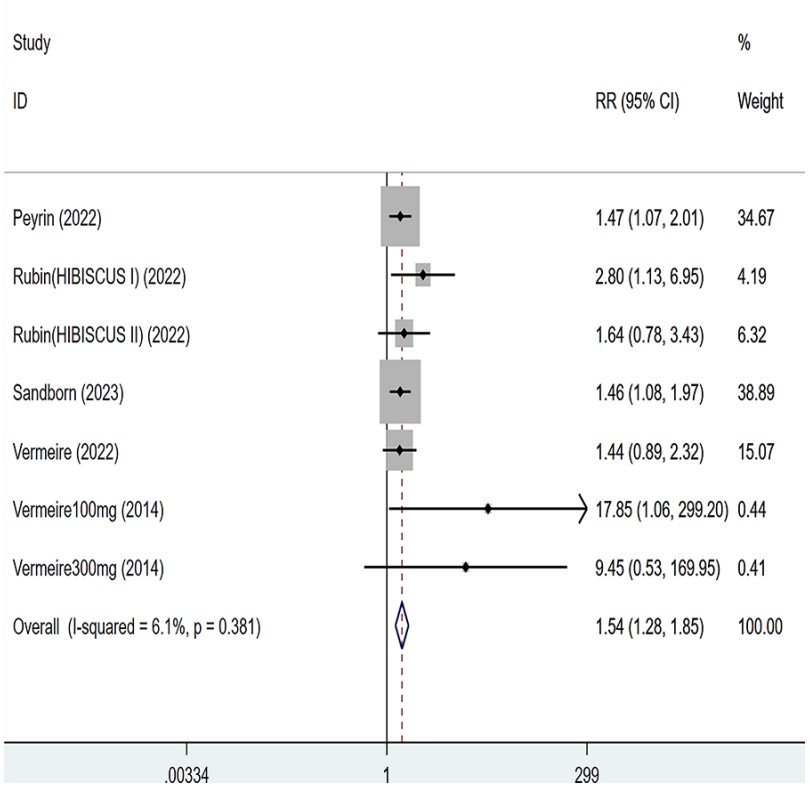

**Figure 2** Forest plot of meta-analysis of clinical response.

data were analyzed using fixed effect model. The results (Fig. 7) suggested that etrolizumab was not statistically significant for serious adverse events compared to placebo (RR = 0.94, 95% CI (0.68–1.31]).

## Histological remission

Three articles were divided into five trials mentioning histological remission and the test of heterogeneity was (I2 = 43.4%, $P = 0.151$), so the data were analyzed by using the fixed effect model and the analysis results (Fig. 8) suggested that compared with placebo, etrolizumab was able to improve IBD patients' histological remission (RR = 1.62, 95% CI [1.26–2.08]), and the difference was statistically significant.

## Published bias

Publication bias was assessed by an Egger's test for clinical remission, clinical response, endoscopic remission, histologic–endoscopic mucosal improvement, adverse events. Which showed no publication bias (Figures S3–S8) for clinical remission ($P = 0.435$), endoscopic improvement ($p = 0.095$), adverse events ($P = 0.937$), histological remission ($P = 0.230$), However, publication bias was detected in clinical response ($P = 0.003$) and endoscopic remission ($P = 0.001$).

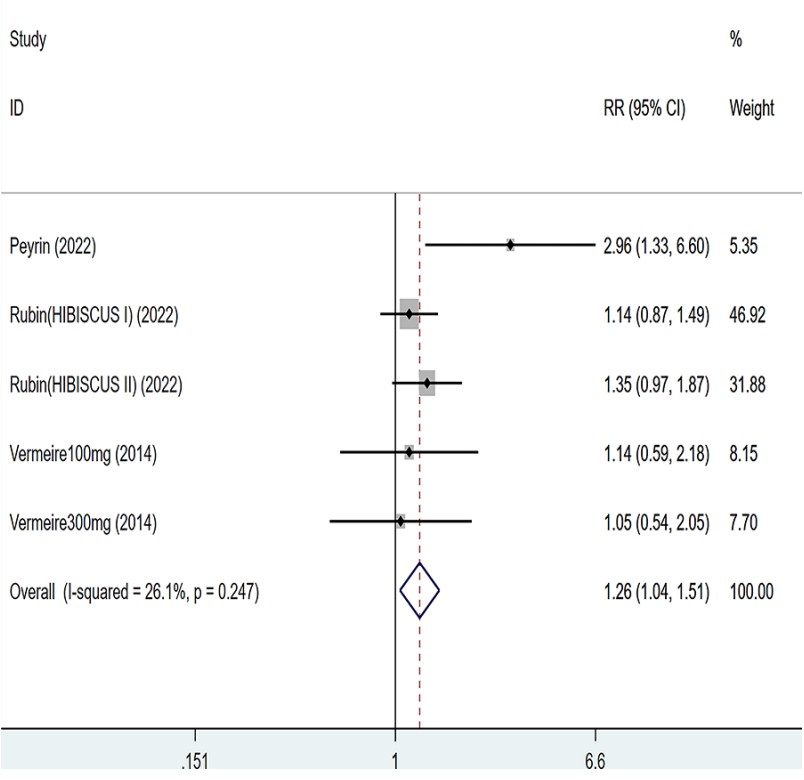

**Figure 3** Forest plot of meta-analysis of clinical remission.

## DISCUSSION

As far as we are concerned, this is not the first time etrolizumab treatment for IBD has been evaluated, but a previous Cochrane study (*Rosenfeld et al., 2015*) included only two original studies. A 2019 meta-analysis (*Motaghi, Ghasemi-Pirbaluti & Zabihi, 2019*) compared an indirect comparison of etrolizumab and infliximab. This is not consistent with our current meta-analysis inclusion metrics. Therefore, the current study included more high-quality studies and the conclusions are more credible.

In this study, we found that etrolizumab improved clinical response, clinical remission, endoscopic remission, endoscopic improvement, and histological remission in patients with IBD without increasing adverse events. The results of our study are further supported by the finding of *Rutgeerts et al. (2013)*. That etrolizumab is safe and well tolerated in patients with moderately to severely active UC. Consistent with these *in vitro* data, preclinical rodent studies demonstrated that blockade of β7 integrins prevented T-cell recruitment to the inflammatory colon in a mouse model of IBD, whereas in a mouse model of multiple sclerosis, blockade of β7 integrins had no effect on lymphocyte homing to the brain. The anti-α4β7 antibody vedolizumab (vedolizumab) has also demonstrated clinical efficacy in UC and CD (*Feagan et al., 2008*; *Feagan et al., 2005*). Like vedolizumab, etrolizumab binds to α4β7, but is unique in that it also blocks the binding of αEβ7 to its ligand,

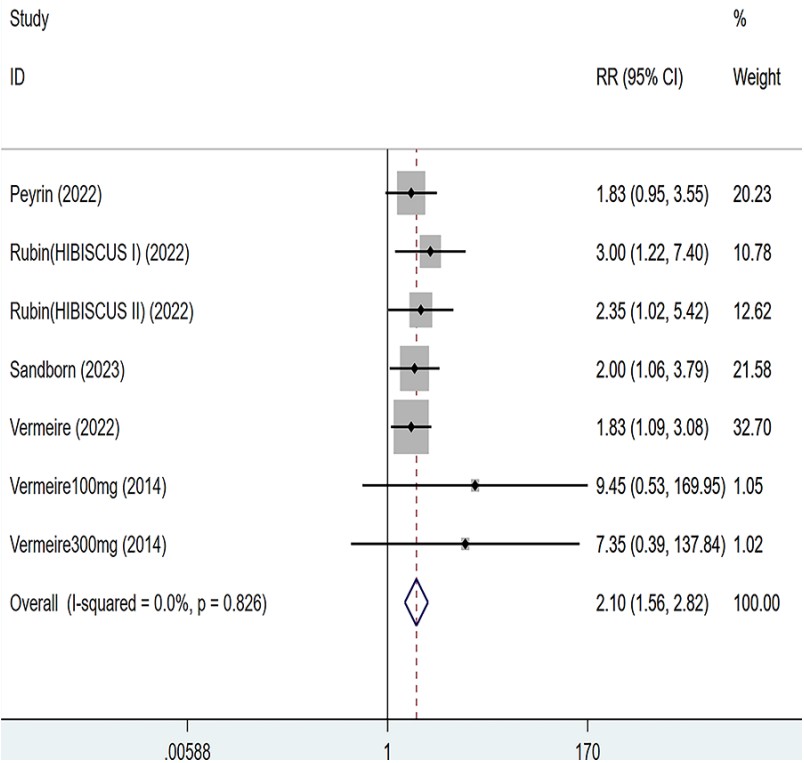

**Figure 4   Forest plot of meta-analysis of endoscopic remission.**

e-calmodulin (*Tang et al., 2018*). Thus, etrolizumab can affect leukocyte composition within the intestinal mucosa through several mechanisms. First, as previously described, it can block entry of α4β7-expressing leukocytes into the intestine by inhibiting extravasation of mucosal endothelial microveins expressing madcam-1 (*Solitano et al., 2021*; *Verstockt et al., 2018*). Notably, MAdCAM-1 expression is increased in patients with UC and CD. In animal models, anti-β7 or α4β7 antibodies are effective in blocking lymphocyte migration into the inflamed intestinal mucosa (*Pérez-Jeldres et al., 2019*). Second, by inhibiting the interaction of αEβ7 with E-cadherin, αEβ7 can directly affect the retention of leukocytes in the intestinal mucosa (*Misselwitz et al., 2020*). αE integrins are expressed at very low levels in peripheral blood, and are found predominantly on intestinal resident cells, including intraepithelial lymphocytes44 and dendritic cells (*Jaensson et al., 2008*). In the lamina propria of the human intestinal mucosa, more than 90% of intraepithelial lymphocytes and 50% of T cells expressed αE β7 integrin, suggesting a unique role in mucosal immunity. In addition, αEβ7 is expressed on intestinal dendritic cells, which are associated with the production of pro-intestinal effector T cells (*Johansson-Lindbom et al., 2005*). Importantly, intraepithelial lymphocytes may exhibit cytotoxic activity against epithelial cells, and cells expressing αEβ7 integrins have been shown to be pathogenic in mouse models of colitis and acute graft-versus-host disease (*Sandborn, 2012*). Etolizumab does not increase adverse

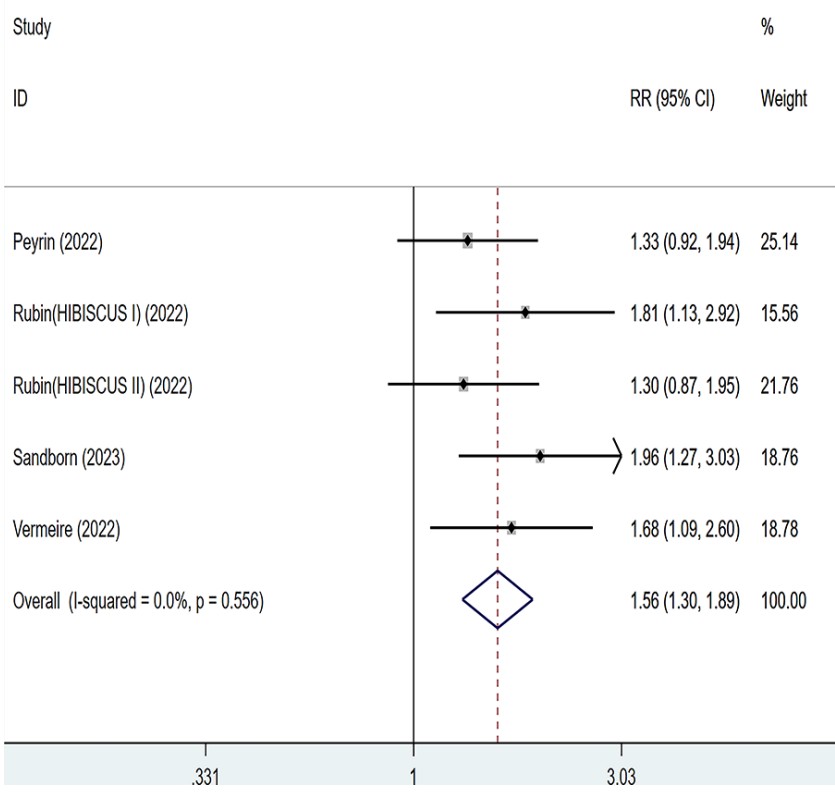

**Figure 5   Forest plot of meta-analysis of endoscopic improvement.**

reactions. This may be because etrolizumab is a monoclonal antibody that acts on the α4β7: MAdCAM-1 and α Eβ7, Ecadherin complexes, which are primarily found in the intestinal epithelium (*Ungar & Kopylov, 2016*). The most common adverse reactions to etolizumab include exacerbation of ulcerative colitis, headache, nausea, abdominal pain, dizziness, malaise, nasopharyngitis, arthralgia, and urinary tract infections (*Gubatan et al., 2021*). Serious adverse reactions to etolizumab include bacterial peritonitis and worsening of ulcerative colitis. Although etolizumab has demonstrated an acceptable safety profile in previous trials, its safety could not be evaluated due to the small number of patients tested (*Weisshof et al., 2018*). Although etrolizumab demonstrated an acceptable safety profile in previous trials, rare adverse events could not be observed due to the small number of patients tested. In addition, due to the small sample size of the trial, we were only able to detect a main effect of etrolizumab. It is expected that additional data from large-volume centers or population studies will provide more information on the safety and efficacy of etrolizumab (*Zundler et al., 2017*).

The risk of bias was low for all included studies. However, the GRADE analysis showed that the overall quality of evidence from etrolizumab trials was moderate or low due to small sample sizes. This means that further trials may change the estimates and improve their accuracy.

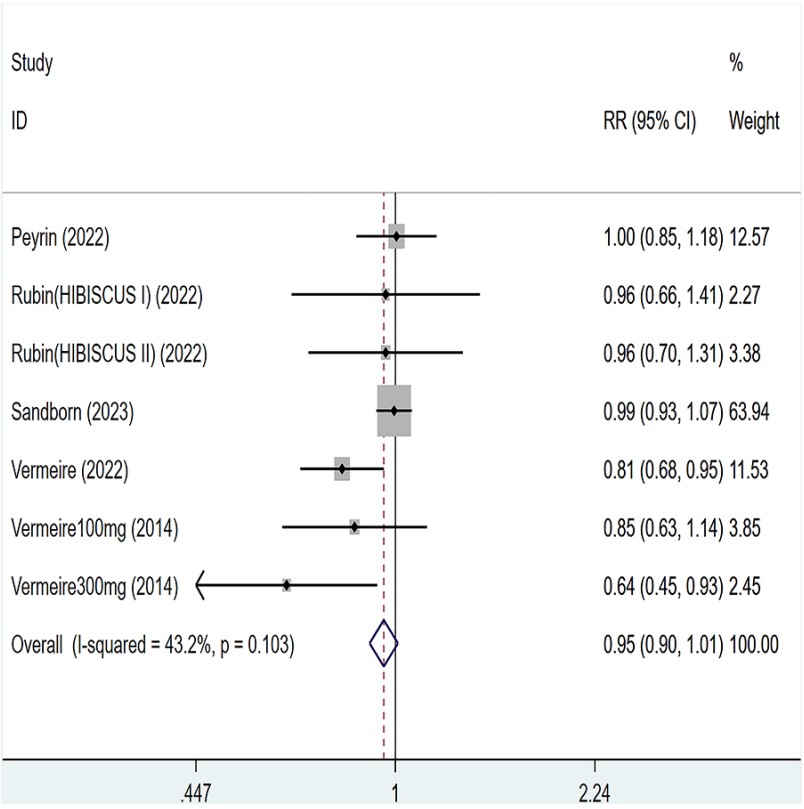

**Figure 6** Forest plot of meta-analysis of any adverse events.

Our study still has several limitations: firstly, due to the small number of studies we included, the number of people involved was small, which may affect the extrapolation of our findings. Secondly, due to the limitation of the number of included studies, we were not able to perform subgroup analyses for outcomes with large heterogeneity. Finally, the dose and duration of time used for etrolizumab were also inconsistent, which may also contribute to the source of heterogeneity.

## CONCLUSION

According to our current study, etrolizumab is a promising drug in IBD, but due to the limitations of the study, we look forward to more high-quality, multicenter, large sample, randomized controlled studies to further support our view.

### Funding
The authors received no funding for this work.

### Competing Interests
The authors declare there are no competing interests.

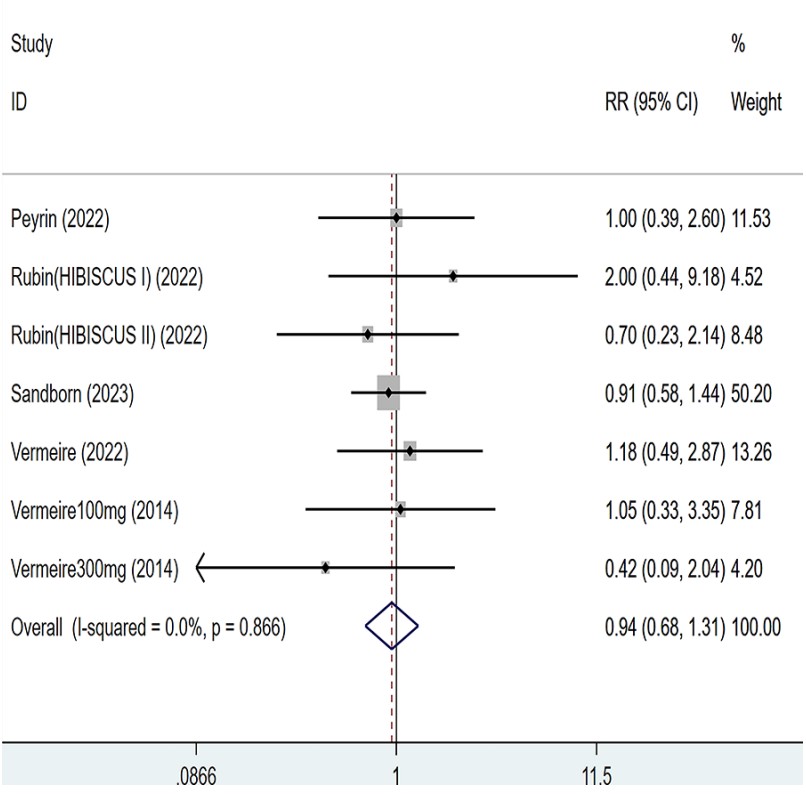

**Figure 7** Forest plot of meta-analysis of serious adverse events.

## Author Contributions

- Yong gang Dai conceived and designed the experiments, performed the experiments, analyzed the data, prepared figures and/or tables, authored or reviewed drafts of the article, and approved the final draft.
- Dajuan Sun conceived and designed the experiments, performed the experiments, analyzed the data, prepared figures and/or tables, authored or reviewed drafts of the article, and approved the final draft.
- Jiahui Liu conceived and designed the experiments, performed the experiments, analyzed the data, prepared figures and/or tables, authored or reviewed drafts of the article, and approved the final draft.
- Xiunan Wei conceived and designed the experiments, performed the experiments, analyzed the data, prepared figures and/or tables, authored or reviewed drafts of the article, and approved the final draft.
- Lili Chi conceived and designed the experiments, performed the experiments, analyzed the data, prepared figures and/or tables, authored or reviewed drafts of the article, and approved the final draft.

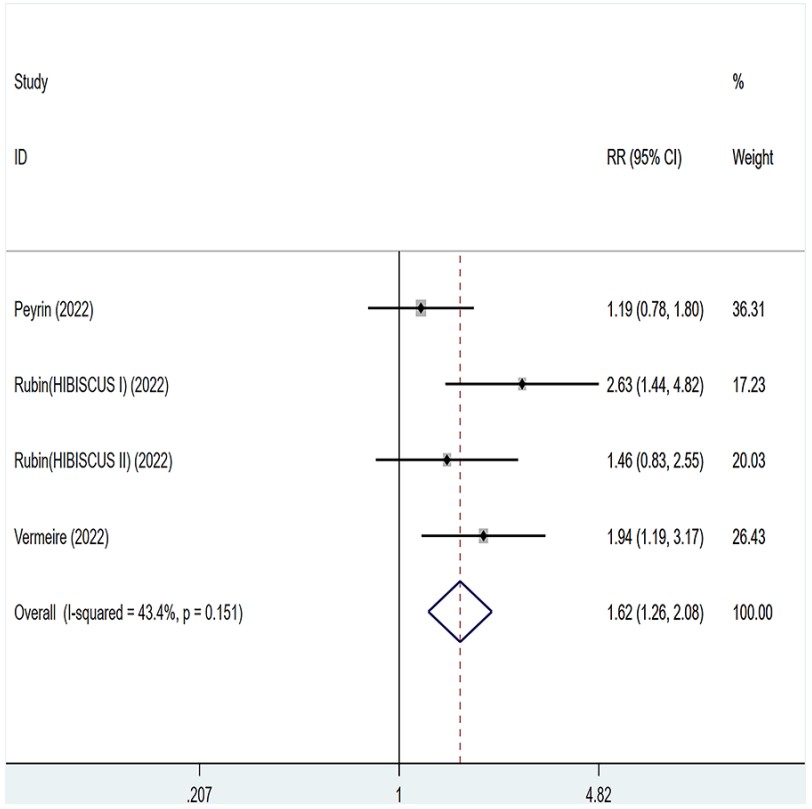

**Figure 8** Forest plot of meta-analysis of histological remission.

- Hongya Wang conceived and designed the experiments, performed the experiments, analyzed the data, prepared figures and/or tables, authored or reviewed drafts of the article, and approved the final draft.

## Data Availability

This is a systematic review / meta-analysis.

## Supplemental Information

Supplemental information for this article can be found online at http://dx.doi.org/10.7717/peerj.17945#supplemental-information.

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
