# Peer review of "Efficacy and safety of etrolizumab in the treatment of inflammatory bowel disease: a meta-analysis"

_PeerJ, doi:10.7717/peerj.17945_

## Round 0.1 · original submission · Major Revisions

If you feel you can revise your manuscript according to the reviewers' comments, please revise your manuscript and submit it. Please also send us your written responses to each of the reviewers' comments.

Yours,

Yoshi

Prof. Yoshinori Marunaka, M.D., Ph.D.

Reviewer 1 ·

Basic reporting

In the study “Efficacy and safety of etrolizumab in the treatment of inflammatory bowel disease: a meta-analysis”, the authors searched published literatures for the randomized controlled trails on etrolizumab for treating IBD and carried out meta-analysis on these studies to assess the efficacy and safety of etrolizumab. The paper is well written with the methods and results clearly demonstrated.

Experimental design

Overall, the methods were well described, however, there are a few points to address.

1. The motivation of the study is to resolve controversies around etrolizumab, please clarify on this point, especially that for the five studies included in the meta-analysis, there were low discrepancies among these studies for metrics such as clinical response and endoscopic improvement.

2. Please include more demographic characteristics from the studies (Table 1), such as race, disease severity, co-morbidity, other medications in use if applicable. Please clarify if there are any heterogeneity among the cohorts from the five studies and whether/how co-variates were considered.

Validity of the findings

The conclusions are well stated and supported by the results. The statistical analyses were carried out with rigor.

Reviewer 2 ·

Basic reporting

Line 25: The text ". 0.4, 1.510]" seems to be redundant here.

Experimental design

Please specify what you mean by “clinical response”. Ensure that each study has used the same definition of clinical response. I recommended you clarify the exact criteria or measures used to define clinical response in each of the included studies, and provide a summary of these criteria in the methods section.

While both ulcerative colitis (UC) and Crohn's disease (CD) fall under the umbrella of inflammatory bowel disease, they do have differences. Please clarify why you could combine the trials for UC and CD in the manuscript. Explain if the therapeutic effects of etrolizumab are expected to be similar across these conditions, or justify the combination based on the similarities in pathophysiology or treatment response.

Validity of the findings

In Figure 1, provide more detailed reasons for the exclusion of 438 out of 445 papers after reading the title and abstract. Simply stating that papers were excluded without providing detailed reasons can be perceived as lacking transparency. It is recommended to categorize the reasons for exclusion and provide a summary in Figure 1. Common reasons for exclusion might include: Irrelevance to the research question; Study type not being a randomized controlled trial (RCT); Duplicates not identified earlier; Focus on a different population or condition; Insufficient outcome data, etc.

A final number of 5 papers is relatively small for a meta-analysis. You should acknowledge it in your limitation section. It is recommended to check the references of the 7 selected papers to ensure no relevant studies were missed. Reference checking can help identify additional studies that may not have been captured in the initial search, increasing the comprehensiveness of the meta-analysis.

Figure 2: The results for Vermeire 100mg (2014) and Vermeire 300mg (2014) show highly variable and unstable RR values, with upper confidence intervals reported as 299.20 and 169.95. Additionally, the overall I² value is very low at 6%. You should Investigate and explain why these specific studies show such extreme and unstable RR values. And please double-check the calculations for the I² value. The significant differences between the RRs suggest that the heterogeneity might be higher than reported.

---

## Round 0.2 · Minor Revisions

Dear Dr. Dai,

Please submit your manuscript revised according to the comments by the reviewer.

Yours,

Yoshi

Prof. Yoshinori Marunaka, M.D., Ph.D.

Reviewer 1 ·

Basic reporting

The paper is well written after addressing reviewers' comments.

Experimental design

The experimental design, methods and investigation are well performed and written.

Validity of the findings

The conclusions are well stated and supported by the results. The statistical analyses were carried out with rigor.

Reviewer 2 ·

Basic reporting

The study is within the scope of the journal, presenting original primary research through a meta-analysis of etrolizumab's efficacy and safety in treating inflammatory bowel disease. The manuscript is generally clear and professionally written. The article follows a professional structure typical for meta-analyses. Figures and tables are relevant and well-labeled. Below are some comments on the basic reporting:

Line 76: Please spell out "MCS".

Figure 1: Indent the sub-items under “Records excluded after reading the title and abstract” with dashes ("-") to clearly indicate that the three reasons listed belong to this main item.

Experimental design

The research question is well-defined, relevant, and meaningful, addressing a significant gap in understanding etrolizumab's clinical utility. The methods section is detailed and allows for replication of the study, with clearly described inclusion and exclusion criteria, data extraction, and analysis methods.

Validity of the findings

Below are one minor comment regarding the validity of the findings:

Figures 4, 5, and 7: Please double-check these figures, as the I-squared values in those plots are reported as 0.0%.

---

## Round 0.3 · accepted · Accept

Dear Dr. Dai,

Congratulations again, and thank you for your submission.

Warm regards,

Yoshi

Prof. Yoshinori Marunaka, M.D., Ph.D.

Reviewer 2 ·

Basic reporting

The authors have addressed all my previous comments regarding basic reporting. The paper is well-written, and I have no further comments.

Experimental design

The authors have addressed all my previous comments regarding experimental design. No further comments.

Validity of the findings

The authors have addressed all my previous comments regarding validity of the findings. No further comments.